# SGD BATCH SATURATION FOR TRAINING WIDE NEURAL NETWORKS

## ABSTRACT

The performance of the mini-batch stochastic gradient method strongly depends on the batch-size that is used. In the classical convex setting with interpolation, prior work showed that increasing the batch size linearly increases the convergence speed, but only up to a point; when the batch size is larger than a certain threshold (the critical batchsize), further increasing the batch size only leads to negligible improvement. The goal of this work is to investigate the relationship between the batchsize and convergence speed for a broader class of nonconvex problems. Building on recent improved convergence guarantees for SGD, we prove that a similar linear scaling and batch-size saturation phenomenon occurs for training sufficiently wide neural networks. We conduct a number of numerical experiments on benchmark datasets, which corroborate our findings.

## 1 INTRODUCTION

Minibatching reduces the number of steps for the stochastic gradient method (SGD) to convergence since it decreases the variance of the stochastic gradient estimator. Practical implementations of SGD exploit this reduction together with parallel computation of gradients to reduce the total wall-clock time to convergence. While batching initially offers a linear reduction in the iteration complexity, an extensively documented experimental observation is that the improvement brought by minibatching saturates after a certain "critical batch size" for models that nearly interpolate the data Golmant et al. (2018); Shallue et al. (2018); see Figure 1 for a numerical illustration. Saturation has also been rigorously proven both for quadratic losses Ma et al. (2018); Zhang et al. (2019), convex losses Woodworth & Srebro (2021) (for a variant of SGD), and certain classes of nonconvex losses Yin et al. (2018); Chen et al. (2018); Gower et al. (2019; 2021) under interpolation assumptions.

While the aforementioned theoretical studies are promising, they are not applicable when training wide neural networks. This work aims to explain why the performance of minibatch SGD with a large stepsize saturates after a certain critical batch size on wide neural networks. We will prove the following theorem, stated here informally for the sake of motivation.

**Theorem 1.1** (Informal)**.** *Consider training a feedforward neural network $f(w, x)$ with width $m$ and linear output layer using SGD with batchsize $b$. Then for sufficiently large $m > 0$, with high probability over initialization $w_0$, the iteration complexity to reach an $\epsilon$-optimal solution scales as*

$$\frac{\max\limits_{i=1,\ldots,n} \|\nabla f(w_0, x_i)\|^2/b + \|K(w_0)\|_{\mathrm{op}} + O\left(\frac{1}{m}\right)}{\lambda_{\min}(K(w_0))} \cdot \log\left(\frac{c}{\varepsilon}\right),$$

*where $K(w_0)$ is the Neural Tangent Kernel (NTK) at initialization and $m$ is the width of the network.*

Thus the theorem shows that the iteration complexity of minibatch SGD exhibits a linear scaling in $b$ roughly up to the critical batch size $b^* = \max\limits_{i=1,\ldots,n} \|\nabla f(w_0, x_i)\|^2/\|K(w_0)\|_{\mathrm{op}}$, after which point increasing the batchsize only leads to negligible improvement. Importantly, past the critical batchsize, the iteration complexity of minibatch SGD matches that of the full-batch gradient method.

Let us briefly explain why the aforementioned works are inapplicable for analyzing training guaranties of wide neural networks—a nonconvex problem in general. The works Yin et al. (2018); Chen et al. (2018), for example, introduce a critical batch size and prove a sublinear convergence

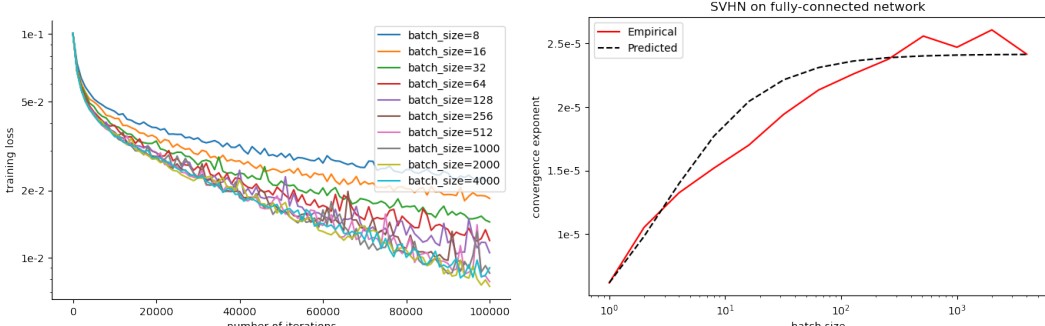

Figure 1: SVHN on fully-connected neural network with MSE loss. The network has 3 hidden layers, with 2000 neurons in each. 100k iterations. Left: the training loss curves vs. batch size. Right: Convergence exponent $h(b)$ and "predicted" curve $\tilde{h}(b)$ (defined in Sec. 5) vs. batch size $b$.

bound for the minibatch SGD under a global Polyak-Łojasiewicz condition (PL) Poljak (1963); Lo-jasiewicz (1963) with a small stepsize that depends inversely on a certain "condition number." In contrast, in the settings of wide-neural networks, it is known that minibatch SGD converges linearly, the PL condition holds locally, and one may, in fact, use a much larger stepsize, depending only on the level of smoothness of the objective Liu et al. (2022; 2023). Thus, the guarantees of Yin et al. (2018); Chen et al. (2018) are inapplicable. Other works analyze minibatch SGD for interpolation problems both under the PL condition Gower et al. (2021) and a "quasi strong convexity" assumption Gower et al. (2019)—a setting where minibatch SGD is known to converge linearly. As in Yin et al. (2018), the work Gower et al. (2021) requires a small stepsize inversely proportional to a "condition number" and suggests the optimal minibatch size is proportional to the size of the training set, which is not true experimentally and does not match the corresponding theoretical behavior in the quadratic or convex setting Ma et al. (2018); Zhang et al. (2019); Woodworth & Srebro (2021). On the other hand, it is known that quasi-strong convexity Gower et al. (2019) is never satisfied for wide neural networks since it entails a locally unique solution.

## 2 MAIN ASSUMPTIONS AND KNOWN RESULTS

The guarantees proved in this work apply to nonlinear least squares problems, with wide neural networks as the primary example. More specifically, throughout this work, we consider the problem:

$$\min_{w \in \mathbb{R}^d} \mathcal{L}(w) = \frac{1}{2n} \sum_{i=1}^{n} (f_i(w) - y_i)^2 = \frac{1}{2} \|F(w)\|^2, \tag{2.1}$$

where $f_i \colon \mathbb{R}^d \to \mathbb{R}$ are some differentiable functions and $y \in \mathbb{R}^n$ is a fixed vector. We will work in the interpolation regime, as summarized in the following assumption.

**Assumption 1** (Interpolation). There exists some point $\bar{w}$ satisfying $f_i(\bar{w}) = y_i$ for all $i = 1, \ldots, n$.

When $n$ is large, the standard procedure for solving the problem 2.1 is the minibatch stochastic gradient method (SGD). In each iteration, the algorithm uniformly samples a batch of indices $S \subset \{1, \ldots, n\}$ of a fixed size $m := |S|$ and performs the update

$$w_{t+1} = w_t - \eta \cdot g(w_t; S) \qquad \text{where} \qquad g(w_t; S) = \frac{1}{|S|} \sum_{i \in S} (f_i(w_t) - y_i) \nabla f_i(w_t).$$

The vector $g(w, S)$ is called the stochastic gradient estimator. The rate of convergence of minibatch SGD is strongly influenced by the second moment of the stochastic gradient estimator $g(w, S)$. Namely, a typical assumption is that there exists a constant $\beta > 0$ such that the estimate

$$\mathbb{E}_S \|g(w, S)\|^2 \le 2\beta \cdot \mathcal{L}(w), \tag{2.2}$$

holds for all $w$ in some ball $B_r(w_0)$. From a high level, much of the paper will be devoted to estimating $\beta$ in terms of $|S|$ under a number of assumptions.

In addition to the interpolation Assumption 1, we impose the following assumption throughout the work. To simplify notation, we let $S$ denote the set of interpolating solutions

$$S \triangleq \arg\min_w \mathcal{L}(w).$$

**Assumption 2.** Fix a point $w_0 \in \mathbb{R}^d$ and parameters $r, \alpha, L > 0$ satisfying the following.

1. **(Existence)** The ball $B_r(w_0)$ intersects the solution set $S$.

2. **(Quadratic growth)** The estimate holds:

$$\mathcal{L}(w) \geq \tfrac{\alpha}{2} \cdot \mathrm{dist}^2(w, S) \qquad \forall w \in B_r(w_0). \tag{2.3}$$

3. **(Lipschitz)** The gradient of each loss function $\nabla f_i$ is $L$-Lipschitz continuous on $B_{2r}(w_0)$.

We now review convergence guarantees for SGD based on these two assumptions, recently developed in (Liu et al., 2023, Theorems 2.5, 3.1). Specifically we will focus on the problem of nonlinear least squares 2.1 where the the Lipschitz constant of $\nabla f_i$ is small. This setting nicely models training of wide neural networks, as we will explain in Section 4.

**Theorem 2.1** (Convergence of minibatch SGD; Liu et al. (2023)). *Suppose that Assumptions 1 and 2 hold, the estimate 2.2 holds for all $w \in B_r(w_0)$, and that $L$ is small in the sense that $L \leq \frac{\alpha}{16r\sqrt{\beta}}$. Fix constants $\delta_1 \in (0, \frac{1}{3})$ and $\delta_2 \in (0, 1)$, and assume $\mathrm{dist}^2(w_0, S) \leq \delta_1^2 r^2$. Consider applying minibatch SGD with stepsize $\eta = \frac{1}{2\beta}$. Then with probability at least $1 - 5\delta_1 - \delta_2$, the estimate $\mathrm{dist}^2(w_t, S) \leq \varepsilon \cdot \mathrm{dist}^2(w_0, S)$ holds after*

$$t \geq \frac{4\beta}{\alpha} \log\left(\frac{1}{\varepsilon\delta_2}\right) \qquad \text{iterations.}$$

In the next section we estimate the value $\beta$ for nonlinear least squares problems (Theorem 3.3). Combining this estimate with Theorem 2.1, we will obtain in Theorem 4.1 scaling laws for how the iteration complexity of minibatch SGD depends on the selected batchsize.

## 3 ESTIMATING $\beta$ AND THE CRITICAL BATCH SIZE FOR SGD

We begin with the following lemma that decomposes the second moment $\mathbb{E}_S \|g(w, S)\|^2$ into a sum of two terms—the first decays linearly in $|S|$ and the second is the squared norm of $\nabla\mathcal{L}(w)$.

**Lemma 3.1** (Decomposition of the second moment). *The inequality holds:*

$$\mathbb{E}_S \|g(w, S)\|^2 \leq \frac{1}{|S|}\mathbb{E}_i[(f_i(w) - y_i)^2 \|\nabla f_i(w)\|^2] + \|\nabla\mathcal{L}(w)\|^2.$$

*Proof.* Let $1_{i \in S}$ and $1_{i,j \in S}$ denote the indicator functions of the events $\{i \in S\}$ and $\{i, j \in S\}$, respectively. We then successively deduce

$$\mathbb{E}_S \|g(w, S)\|^2 = \mathbb{E}_S \left\| \frac{1}{|S|} \sum_{i \in S} (f_i(w) - y_i)\nabla f_i(w) \right\|^2$$

$$= \frac{1}{|S|^2}\mathbb{E}_S \left\| \sum_{i=1}^n (f_i(w) - y_i)\nabla f_i(w) 1_{i \in S} \right\|^2$$

$$= \frac{1}{|S|^2} \sum_{i=1}^n (f_i(w) - y_i)^2 \|\nabla f_i(w)\|^2 P(i \in S)$$

$$+ \frac{1}{|S|^2} \sum_{i \neq j} (f_i(w) - y_i)(f_j(w) - y_j)\langle \nabla f_i(w), \nabla f_j(w) \rangle P(i, j \in S),$$

where the last inequality follows from expanding the square and using linearity of expectation. A simple computation shows that $P(i \in S) = (1 - (1 - \frac{1}{n})^{|S|}) \leq \frac{|S|}{n}$, where the last estimate follows

from Bernoulli's inequality. Similarly for $i \neq j$, we compute $P(i, j \in S) = P(i \in S \mid j \in S)P(j \in S) \leq \frac{|S|-1}{n-1} \cdot \frac{|S|}{n} \leq \frac{|S|^2}{n^2}$. Plugging this back into the equation and recognizing that the last term is bounded by $\|\nabla F(w)^\top F(w)\|^2$ completes the proof. $\qquad\square$

Next, we estimate $\beta$ for the problem of linear least squares, which slightly sharpens the analogous result in Ma et al. (2018). We include it here as motivation for general nonlinear least squares 2.1.

**Theorem 3.2** (Linear least squares). *Consider the problem 2.1 in the setting where $f_i(w) = x_i^\top w$ for some vectors $x_i \in \mathbb{R}^d$ satisfying the moment bound:*

$$\mathbb{E}_i \|x_i\|^2 x_i x_i^\top \preceq \gamma \cdot \mathbb{E}_i x_i x_i^\top. \tag{3.1}$$

*Suppose moreover that we are in the interpolation regime, that is there exists some $\bar{w}$ satisfying $F(\bar{w}) = 0$. Then equation 2.2 holds for all $w$ with*

$$\beta = \frac{\gamma}{|S|} + \frac{1}{n}\|X\|_{\mathrm{op}}^2,$$

*where $X$ denotes the matrix having $x_i$ as its rows.*

*Proof.* Define the displacement vector $v = w - \bar{w}$ and note that we may write $f_i(w) - y_i = \langle x_i, v \rangle$. Therefore applying Lemma 3.1 we obtain

$$\mathbb{E}_S \|g(w, S)\|^2 \leq \frac{1}{|S|} v^\top \mathbb{E}_i[\|x_i\|^2 x_i x_i^\top] v + \left\| \frac{1}{n} X^\top X v \right\|^2$$

$$\leq \frac{\gamma}{|S|} v^\top \mathbb{E}_i[x_i x_i^\top] v + \frac{1}{n^2} \|X\|_{\mathrm{op}}^2 \|Xv\|^2$$

$$= \left( \frac{2\gamma}{|S|} + \frac{2}{n}\|X\|_{\mathrm{op}}^2 \right) \cdot \mathcal{L}(w),$$

thereby completing the proof. $\qquad\square$

A few comments are in order. First, the condition 3.1 roughly stipulates that the second-order moment of $x_i$ is bounded by a multiple of the first-order moment. Conditions of this type have been used extensively in the literature, such as Ma et al. (2018); Dieuleveut et al. (2017); Jain et al. (2018). In particular, 3.1 holds automatically with $\gamma = \max_{i=1,\ldots,n} \|x_i\|^2$, and this choice is optimal if $x_i$ are pairwise orthogonal—often a good approximation in the regime of interest $d \gg n$. Conversely taking the trace of both sides of 3.1 shows that any valid $\gamma$ must be larger than $\frac{1}{n}\sum_{i=1}^n \|x_i\|^2$. Theorem 3.2 establishes a linear scaling of $\beta$ in the batchsize up to the critical batchsize

$$b^* = \frac{\gamma}{\frac{1}{n}\|X\|_{\mathrm{op}}^2},$$

past which point $\beta$ becomes nearly identical to $\frac{1}{n}\|X\|_{\mathrm{op}}^2$—which is exactly is equal to the optimal choice of $\beta$ for the full-batch gradient $S = \{1, \ldots, n\}$. Next, we extend Theorem 3.2 to the setting of nonlinear least squares 2.1 in the case when the gradient of each function $f_i(w)$ has a small Lipschitz constant. The reader should keep in mind the direct parallel with the linear case (Theorem 3.2).

**Theorem 3.3** (Nonlinear least squares). *Suppose that Assumptions 1 and 2 hold and that for some $\gamma > 0$ the random vector $x_i := \nabla f_i(w_0)$ satisfies the moment bound*

$$\mathbb{E}_i \|x_i\|^2 x_i x_i^\top \preceq \gamma \cdot \mathbb{E}_i x_i x_i^\top. \tag{3.2}$$

*Then equation 2.2 holds for all $w \in B_r(w_0)$ with*

$$\beta = \frac{16\gamma + \frac{200\gamma L^2 r^2}{\alpha}}{|S|} + 4L^2 r^2 + 4\|\nabla F(w_0)\|_{\mathrm{op}}^2.$$

*Proof.* Throughout, we let $w \in B_r(w_0)$ be arbitrary. Lemma 3.1 yields the estimate

$$\mathbb{E}_S \|g(w, S)\|^2 \leq \frac{1}{|S|} \underbrace{\mathbb{E}_i[(f_i(w) - y_i)^2 \|\nabla f_i(w)\|^2]}_{=:P_1} + \underbrace{\|\nabla F(w)^\top F(w)\|_2^2}_{P_2}.$$

We may upper bound $P_2$ as

$$P_2 \leq \|\nabla F(w)\|_{\mathrm{op}}^2 \|F(w)\|^2 = 2\|\nabla F(w)\|_{\mathrm{op}}^2 \mathcal{L}(w). \tag{3.3}$$

Note moreover that

$$\|\nabla F(w) - \nabla F(w_0)\|_{\mathrm{op}}^2 \leq \|\nabla F(w) - \nabla F(w_0)\|_F^2$$

$$\leq \frac{1}{n}\sum_{i=1}^n \|\nabla f_i(w) - \nabla f_i(w_0)\|_2^2 \leq L^2\|w - w_0\|^2.$$

Therefore we deduce $P_2 \leq 4(\|\nabla F(w_0)\|_{\mathrm{op}}^2 + L^2\|w - w_0\|^2)\mathcal{L}(w)$. It remains to bound $P_1$. To this end, note $\|\nabla f_i(w)\|^2 \leq 2\|\nabla f_i(w_0)\|^2 + 2L^2\|w - w_0\|^2$. Therefore, we may estimate

$$P_1 \leq 2\mathbb{E}_i(f_i(w) - y_i)^2\|\nabla f_i(w_0)\|^2 + 2L^2 \cdot \mathbb{E}_i(f_i(w) - y_i)^2\|w - w_0\|^2$$

$$= 2\mathbb{E}_i(f_i(w) - y_i)^2\|\nabla f_i(w_0)\|^2 + 4L^2\|w - w_0\|^2 \cdot \mathcal{L}(w). \tag{3.4}$$

Now let $\bar{w}$ denote a closest point to $w$ in $\arg\min\mathcal{L}$ and note that by the triangle inequality $\bar{w}$ lies in $B_{2r}(w_0)$. Using the fundamental theorem of calculus, we may write

$$f_i(w) - y_i = f_i(w) - f_i(\bar{w}) = \int_0^1 \langle \nabla f_i(\bar{w} + t(w - \bar{w})), w - \bar{w}\rangle \ dt \tag{3.5}$$

$$= \langle \nabla f_i(w_0), w - \bar{w}\rangle + E,$$

where $|E| \leq \frac{L}{2}\|w - \bar{w}\|(3\|w_0 - w\| + \|w_0 - \bar{w}\|) \leq \frac{5rL}{2}\|w - \bar{w}\|$. Therefore, we may estimate

$$\mathbb{E}_i(f_i(w) - y_i)^2\|\nabla f_i(w_0)\|^2 \leq 2\mathbb{E}_i\langle \nabla f_i(w_0), w - \bar{w}\rangle^2\|\nabla f_i(w_0)\|^2$$

$$+ 2\mathbb{E}_i\|\nabla f_i(w_0)\|^2 E^2. \tag{3.6}$$

Observe that setting $v = w - \bar{w}$ we may write

$$\mathbb{E}_i\langle \nabla f_i(w_0), w - \bar{w}\rangle^2\|\nabla f_i(w_0)\|^2 = v^\top \left[\mathbb{E}_i\|\nabla f_i(w_0)\|^2\nabla f_i(w_0)\nabla f_i(w_0)^\top\right] v$$

$$\leq \gamma \cdot v^\top \left[\mathbb{E}_i\nabla f_i(w_0)\nabla f_i(w_0)^\top\right] v \tag{3.7}$$

$$= \gamma \cdot \mathbb{E}_i\langle \nabla f_i(w_0), v\rangle^2$$

$$\leq \gamma \cdot \mathbb{E}_i(2(f_i(w) - y_i)^2 + 2E^2) \tag{3.8}$$

$$= 4\gamma\mathcal{L}(w) + 2\gamma E^2, \tag{3.9}$$

where the equation 3.7 follows from equation 3.2 and equation 3.8 follows from equation 3.5. Therefore, combining equations 3.4, 3.6, and 3.9 we conclude

$$P_1 \leq (8\gamma + 4\mathbb{E}_i\|\nabla f_i(w_0)\|^2) \cdot E^2 + (16\gamma + 4L^2\|w - w_0\|^2) \cdot \mathcal{L}(w). \tag{3.10}$$

Next note that upon taking the trace in the definition of $\gamma$, we have $\mathbb{E}_i\|\nabla f_i(w_0)\|^2 \leq \gamma$. Moreover, using the quadratic growth condition, we see that

$$E^2 \leq \frac{25L^2r^2}{4}\|w - \bar{w}\|^2 \leq \frac{50L^2r^2}{4\alpha} \cdot \mathcal{L}(w). \tag{3.11}$$

Combining the estimates 3.10 and 3.11 completes the proof. $\qquad\square$

Theorem 3.3 imposes a number of nontrivial assumptions. First, the gradient of each function $f_i$ has to be Lipschitz continuous with constant $L$. In particular, for the ensuing results to be meaningful, $L$ must be very small; this is the case for wide neural networks as we discuss in Section 4. Second, the theorem imposes the quadratic growth condition 2.3; this again is automatic for wide neural networks. The final assumption equation 3.2 is directly analogous to equation 3.1 in the linear case. In particular, we may set $\gamma = \max_{i=1,\ldots,n}\|\nabla f_i(w_0)\|^2$. Importantly, this quantity is computable because it depends only on gradient of $f_i$ at the center point $w_0$. Under these assumptions, Theorem 3.3 establishes a linear scaling of $\beta$ in the batchsize up to the **critical batchsize**

$$b^* = \frac{4\gamma + \frac{50\gamma L^2r^2}{\alpha}}{\|\nabla F(w_0)\|_{\mathrm{op}}^2 + L^2r^2} \approx \frac{4\gamma}{\|\nabla F(w_0)\|_{\mathrm{op}}^2} \qquad \text{for } L \approx 0.$$

Past this batchsize, $\beta$ becomes nearly identical to $4(\|\nabla F(w_0)\|_{\mathrm{op}}^2 + L^2r^2)$. In particular, observe that $\|\nabla F(w_0)\|_{\mathrm{op}}^2$ is exactly equal to the optimal choice of $\beta$ for the full-batch gradient $S = \{1,\ldots,n\}$ on the linearized problem $\min_w \|F(w_0) + \nabla F(w_0)(w - w_0)\|^2$ at $w_0$.

## 4 CONSEQUENCES FOR NONLINEAR LEAST SQUARES AND WIDE NEURAL NETWORKS.

In particular, combining Theorems 2.1 and Theorem 3.3 yields a precise expression for how the batchsize effects the iteration complexity of minibatch SGD— the content of the following theorem.

**Theorem 4.1** (Batchsize and iteration complexity). *Suppose that Assumptions 1 and 2 hold, and suppose that $L$ is small in the sense that*

$$L \leq \frac{\alpha}{16 r \sqrt{\beta}} \qquad where \qquad \beta \triangleq \frac{16\gamma + \frac{200\gamma L^2 r^2}{\alpha}}{|S|} + 4L^2 r^2 + 4\|\nabla F(w_0)\|_{\mathrm{op}}^2.$$

*Fix constants $\delta_1 \in (0, \frac{1}{3})$ and $\delta_2 \in (0, 1)$, and assume $\mathrm{dist}^2(w_0, S) \leq \delta_1^2 r^2$. Consider applying minibatch SGD with stepsize $\eta = \frac{1}{2\beta}$. Then with probability at least $1 - 5\delta_1 - \delta_2$, the estimate* $\mathrm{dist}^2(w_t, S) \leq \varepsilon \cdot \mathrm{dist}^2(w_0, S)$ *holds after $t \geq \frac{4\beta}{\alpha} \log \left( \frac{1}{\varepsilon \delta_2} \right)$ iterations.*

Thus assuming that $L$ is small and ignoring log factors, the iteration complexity of SGD is roughly

$$\frac{\gamma/\alpha}{|S|} + \frac{\|\nabla F(w_0)\|_{\mathrm{op}}^2}{\alpha} + O(L).$$

Thus we see a linear scaling of the complexity up to the critical batchsize, after which point it roughly coincides with the complexity of solving the problem $\min_w \|F(w_0) + \nabla F(w_0)(w - w_0)\|^2$.

We next discuss consequences of Theorem 4.1 for a nonlinear least squares problem arising from fitting a wide neural network. Setting the stage, an $l$-layer (feedforward) neural network $f(w; x)$, with parameters $w$, input $x$, and linear output layer is defined as follows:

$$\alpha^{(0)} = x,$$
$$\alpha^{(i)} = \sigma \left( \frac{1}{\sqrt{m_{i-1}}} W^{(i)} \alpha^{(i-1)} \right), \quad \forall i = 1, \ldots, l-1$$
$$f(w; x) = \frac{1}{\sqrt{m_{l-1}}} W^{(l)} \alpha^{(l-1)}.$$

Here, $m_i$ is the width (i.e., number of neurons) of $i$-th layer, $\alpha^{(i)} \in \mathbb{R}^{m_i}$ denotes the vector of $i$-th hidden layer neurons, $w := \{W^{(1)}, W^{(2)}, \ldots, W^{(l)}, W^{(l+1)}\}$ denotes the collection of the parameters (or weights) $W^{(i)} \in \mathbb{R}^{m_i \times m_{i-1}}$ of each layer, and $\sigma$ is the activation function, e.g., $sigmoid$, $tanh$, linear activation. We also denote the width of the neural network as $m := \min_{i \in [l]} m_i$, i.e., the minimal width of the hidden layers. The neural network is usually randomly initialized, i.e., each individual parameter is initialized i.i.d. following $\mathcal{N}(0, 1)$. Henceforth, we assume that the activation functions $\sigma$ are twice differentiable, $L_\sigma$-Lipschitz, and $\beta_\sigma$-smooth.

**Remark 4.1.** The order notation $\Omega(\cdot)$ and $O(\cdot)$ will suppress multiplicative factors of polynomials (up to degree $l$) of the constants $C_z$, $L_\sigma$ and $\beta_\sigma$.

Given a dataset $\mathcal{D} = \{(x_i, y_i)\}_{i=1}^n$, we fit the neural network by solving the least squares problem 2.1 with $f_i(w) \triangleq f_i(w, x_i)$. We assume that all the the data inputs $x_i$ are bounded, i.e., $\|x_i\| \leq C$ for some constant $C$. We now aim to show that when the width $m$ is sufficiently large, the assumptions of Theorem 4.1 hold, and consequently deduce the linear scaling plus saturation phenomenon. With this in mind, we review a few basic facts about wide neural networks.

Define the Neural Tangent Kernel $K(w_0) = \nabla F(w_0) \nabla F(w_0)^\top$ at the random initial point $w_0 \sim N(0, I)$ and let $\lambda_0$ be the minimal eigenvalue of $K(w_0)$. The value $\lambda_0$ is positive with high probability in Du et al. (2018; 2019). Namely, under a mild non-degeneracy condition on the data set, the smallest eigenvalue $\lambda_\infty$ of NTK of an infinitely wide neural network is positive (see Theorem 3.1 of Du et al. (2018)). Moreover, if the width satisfies $m = \Omega(\frac{n^2 \cdot 2^{O(l)}}{\lambda_\infty^2} \log \frac{nl}{\epsilon})$, then with probability at least $1 - \epsilon$ the estimate $\lambda_0 > \frac{\lambda_\infty}{2}$ holds (Du et al., 2019, Remark E.7). Of course, this is a worst case bound and for our purposes we only need to ensure that $\lambda_0$ is positive. It will also be important to know that $\|F(w_0)\|^2 = O(1)$, which indeed occurs with high probability as shown in Jacot et al. (2018). To simplify notation, let us lump these two probabilities together and define

$$p \triangleq \mathbb{P}\{\lambda_0 > 0, \|F(w_0)\|^2 \leq C\}.$$

Next, we require the following theorem, which shows that when the width $m$ is sufficiently large, the function $w \mapsto f(w, x)$ is nearly linear on $B_r(w_0)$. This in particular provides an upper bound on $L$, which can be made arbitrarily small by increasing $m$.

**Theorem 4.2** (Transition to linearity Liu et al. (2020) and the PŁ condition Liu et al. (2022)). *Given $r > 0$, with probability $1 - p - 2 \exp(-\frac{ml}{2}) - (1/m)^{\Theta(\ln m)}$ of initialization $w_0 \sim N(0, I)$, it holds:*

$$\|\nabla^2 f(w, x)\|_{\mathrm{op}} = \tilde{O}\left(\frac{r^{3l}}{\sqrt{m}}\right) \qquad \forall w \in B_r(w_0), \ \|x\| \leq C. \tag{4.1}$$

Validity of quadratic growth and the fact that $B_w(w_0)$ intersects $S$ was proved in (Liu et al., 2023, Theorem 3.4); we record this result next.

**Theorem 4.3** (Quadratic growth for wide NNs). *With probability $1 - p - 2 \exp(-\frac{ml}{2}) - (1/m)^{\Theta(\ln m)}$ with respect to the initialization $w_0 \sim N(0, I)$, as long as*

$$m = \tilde{\Omega}\left(\frac{nr^{6l+2}}{\lambda_0^2}\right) \qquad \text{and} \qquad r = \Omega\left(\frac{1}{\sqrt{\lambda_0}}\right),$$

*quadratic growth 2.3 holds on $B_r(w_0)$ with parameter $\lambda_0/2$ and $B_r(w_0) \cap S$ is nonempty.*

Theorems 4.2 and 4.2 directly imply that the assumptions of Theorem 4.1 are valid. A direct application of Theorem 4.1 yields the main result of the section.

**Theorem 4.2** (Minibatch SGD for wide neural networks). *Fix constants $\delta_1 \in (0, \frac{1}{3})$, $\delta_2 \in (0, 1)$, $\varepsilon > 0$ and $t \in \mathbb{N}$. Then with probability $1 - p - \delta_1 - \delta_2 - 2 \exp(-\frac{ml}{2}) - (1/m)^{\Theta(\ln m)}$, as long as*

$$m = \tilde{\Omega}\left(\frac{nr^{6l+2}}{\lambda_0^2}\right) \qquad \text{and} \qquad r = \Omega\left(\frac{1}{\delta_1 \sqrt{\lambda_0}}\right), \tag{4.2}$$

*both Assumptions 1 and 2 hold and minibatch SGD with a constant stepsize $\eta = \frac{1}{\beta}$ finds a point $w_t$ satisfying $\mathrm{dist}^2(w_t, S) \leq \varepsilon \cdot \mathrm{dist}^2(w_0, S)$ after at most after*

$$t \geq \frac{8\beta}{\lambda_0} \log\left(\frac{1}{\varepsilon \delta_2}\right) \qquad \text{iterations,}$$

*where*

$$\beta \triangleq \frac{16 \cdot \max\limits_{i=1,\ldots,n} \|\nabla f(w, x_i)\|^2}{|S|} + 4\|\nabla F(w_0)\|_{\mathrm{op}}^2 + O\left(\frac{r^{6l+2}}{m}\left(1 + \frac{1}{\lambda_0|S|}\right)\right). \tag{4.3}$$

Note that the width requirements in the theorem are nearly identical to those for the full batch gradient descent Liu et al. (2022), with the exception being that the requirement $r = \Omega\left(\frac{1}{\sqrt{\lambda_0}}\right)$ is strengthened to $r = \Omega\left(\frac{1}{\delta_1 \sqrt{\lambda_0}}\right)$. That is, the radius $r$ needs to shrink by the probability of failure. The special case of this theorem with batchsize of $|S| = 1$ appeared in (Liu et al., 2023, Cor. 3.4).

The third term in equation 4.3 is negligible in the regime of interest for $m$ 4.2. Thus, we see that increasing the batchsize linearly decreases the iteration complexity roughly up to the critical batchsize $b^* = \max\limits_{i=1,\ldots,n} \|\nabla f(w, x_i)\|^2 / \|\nabla F(w_0)\|_{\mathrm{op}}^2$, after which point the iteration complexity of SGD matches that of the full-batch gradient method on the linearized problem.

## 5 EXPERIMENTAL RESULTS

In this section, we numerically illustrate the saturation effects in mini-batch SGD when using large batch sizes—a phenomenon extensively explored in Golmant et al. (2018); Shallue et al. (2018). In our experiments, the stepsize is held constant across varying batch sizes $b$. Although the optimal stepsize does depend on $b$, we opt for a fixed stepsize to circumvent extensive hyperparameter tuning.

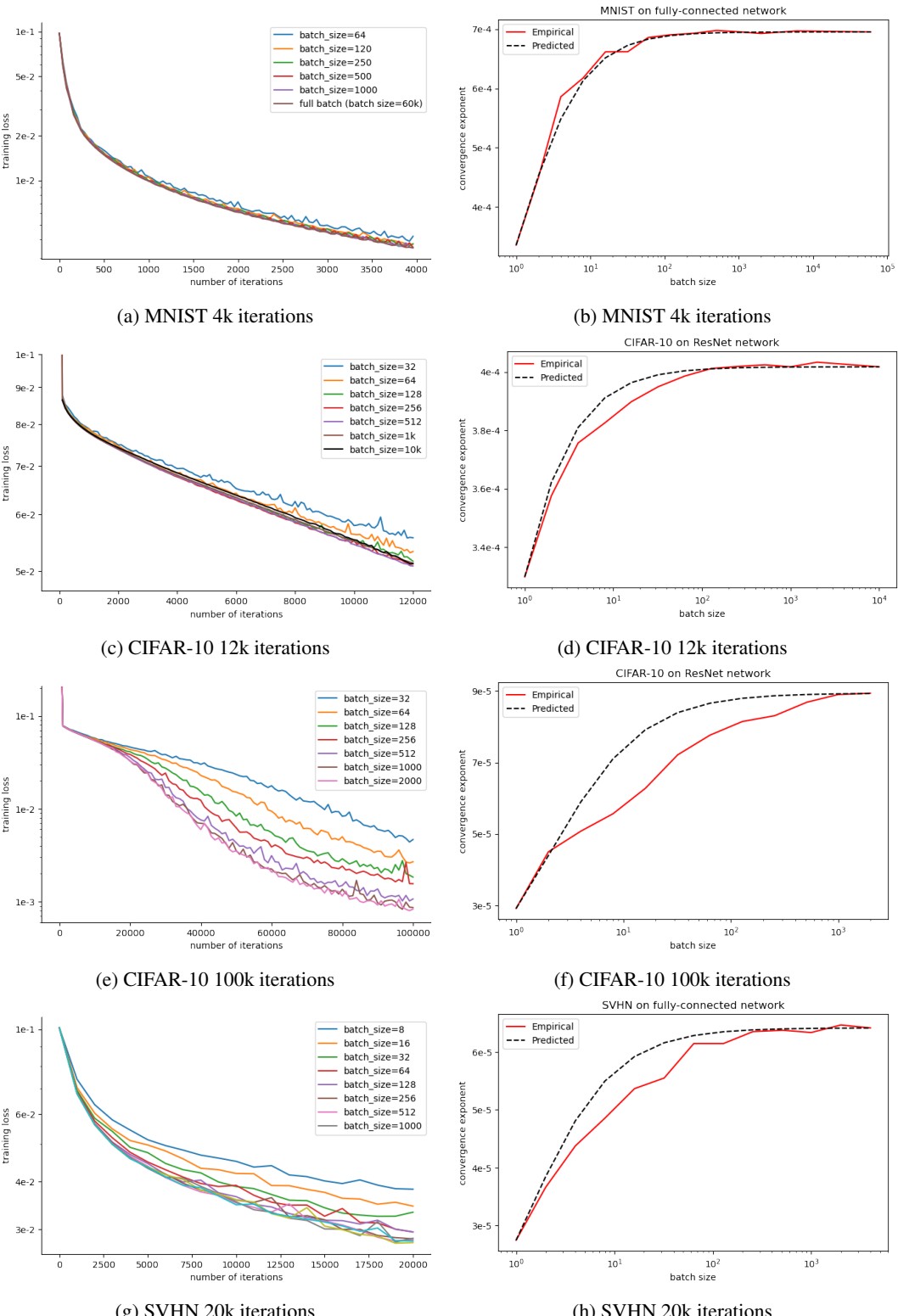

(a) MNIST 4k iterations

(b) MNIST 4k iterations

(c) CIFAR-10 12k iterations

(d) CIFAR-10 12k iterations

(e) CIFAR-10 100k iterations

(f) CIFAR-10 100k iterations

(g) SVHN 20k iterations

(h) SVHN 20k iterations

Figure 2: Left: the training loss vs. batch size. Right: Convergence exponent $h(b)$ vs. batch size $b$.

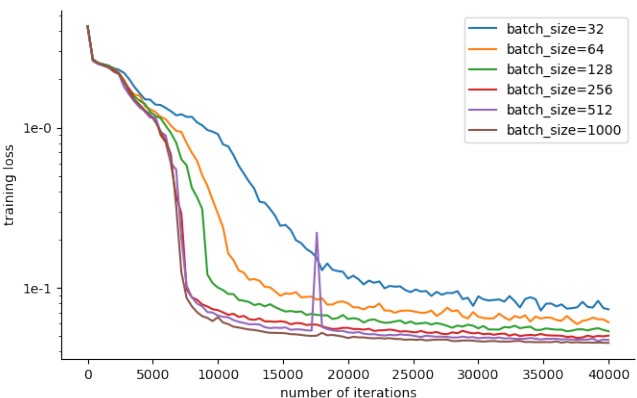

Figure 3: Training loss curves vs. batch size for NanoGPT.

In our experiments we examine the convergence exponent $h(b)$, defined through the expression $\mathcal{L}(w_T) = \mathcal{L}(w_0)\exp(-h(b)T)$ where $T$ is the total number of training iterations. We provide an empirical estimate of $h$ by plotting the function $\tilde{h}(b)$ defined through linearly interpolating $1/h(1)$ and $1/h(n)$ according to the formula $\frac{1}{\tilde{h}(b)} = \frac{1}{b} \cdot \frac{1}{h(1)} + \frac{b-1}{b} \cdot \frac{1}{h(n)}$.

In our experiments, we consider several configurations of neural networks and training durations. For the MNIST dataset, we use a fully-connected NN with 3 hidden layers, each having 1000 neurons. The network is trained for 4k iterations using MSE loss (see Figure 2a). For CIFAR-10, we employ a ResNet-28 architecture and also use MSE loss. Two training durations are considered: 12k iterations (Figure 2c) and 100k iterations (Figure 2e). Finally, for the SVHN dataset, we again use a fully-connected NN but with 3 hidden layers and 2000 neurons in each layer. The network is trained for 20k iterations with MSE loss (Figure 2g). Additionally, we investigate the NanoGPT architecture, a 6-layer Transformer with 6 heads per layer and 384 feature channels. This character-level GPT has a context size of up to 256 characters and is trained on the works of Shakespeare, converted into a continuous string. Each training sample consists of a 256-character substring (Figure 3).

We demonstrate the saturation effect in two ways: First, the iteration-wise training loss curves become closer as the batch size increases; especially for large batch sizes, iteration-wise training loss curves are almost identical. Second, the empirically estimated convergence exponent $h(b)$ aligned well with the theoretically predicted/interpolated $\tilde{h}(b)$, and both curves flatten for large batch sizes.

## 6  CONCLUSION

In this work, we established a quantitative relationship between batch size and iteration complexity for training wide neural networks with SGD. Specifically, we have shown both theoretically and empirically that the iteration complexity scales linearly with batch size up to a critical batch size, after which further increasing the batch size leads to negligible improvements. This scaling behavior aligns with previous results in convex optimization as well as experimental observations in deep learning. Our analysis helps provide a theoretical justification for the common heuristic of choosing the largest batch size that still fits in memory and scales training efficiently. Beyond this critical batch size, diminishing returns are incurred by using larger mini-batches.

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
