# OpenReview forum: "SGD batch saturation for training wide neural networks"
_ICLR.cc/2024/Conference — Submitted to ICLR 2024_

### Official Review · Reviewer_mzrR · 2023-10-15

**Soundness:** 2 fair
**Presentation:** 1 poor
**Contribution:** 2 fair
**Rating:** 1
**Confidence:** 5

**Summary:**

This paper considers the performance of SGD under certain assumptions (Assumptions 1 and 2) for solving a minimization problem (Equation (2.1) that is to minimize the loss function $\mathcal{L}(w)$) in wide neural networks.

The theoretical analyses of this paper are presented in Sections 3 and 4. Section 3 indicates sufficient conditions (Inequalities (3.1) and (3.2)) to show explicitly the form of $\beta$ such that $\mathbb{E}_S \Vert g(w,S) \Vert^2 \leq 2 \beta \mathcal{L}(w)$. Section 4 indicates the number of iterations of SGD using a constant step-size $\eta = \frac{1}{\beta}$ to achieve an $\epsilon$-approximation (defined by $\mathrm{dist}^2(w_t, S) \leq \epsilon \mathrm{dist}^2(w_0, S)$, where $w_t$ is the point generated by SGD and $S$ is the solution set) is more than or equal to $\frac{4 \beta}{\alpha} \log (\frac{1}{\epsilon \delta_2})$, where $\alpha$ is defined as in Assumption 2 (Quadratic growth; Inequality (2.3)) and $\delta_2 \in (0,1)$.

Section 5 gives some numerical results to support the theoretical analyses.

**Strengths:**

The strength of this paper is to consider the performance of SGD under certain assumptions (Assumptions 1 and 2) for solving a minimization problem (Equation (2.1) that is to minimize the loss function $\mathcal{L}(w)$) in wide neural networks. The width $m$ in a wide neural network depends on the setting of $\beta$ (Theorem 4.2). Moreover, Theorem 4.3 indicates conditions of a radius $r$ and $m$ such that Assumption 2 (Quadratic growth; Inequality (2.3)) holds.  Theorem 4.4 (Although the original is Theorem 4.2, I guess the fourth theorem in Section 4) indicates sufficient conditions (Condition (4.2) for $r$ and $m$) to satisfy Assumptions 1 and 2. Theorem 4.4 also shows that SDG achieves an $\epsilon$-approximation after $\frac{8\beta}{\lambda_0} \log (\frac{1}{\epsilon \delta_2})$ iterations, where $\lambda_0 $ is the minimum eigenvalue of the neural tangent kernel.

**Weaknesses:**

The theoretical results in Sections 3 and 4 are interesting. However, the mathematical preliminaries and explanations are insufficient. The abstract indicates that  "The goal of this work is to investigate the relationship between the batchsize and convergence speed for a broader class of nonconvex problems." However, I cannot check any results for the goal. In addition, the abstract indicates that "We conduct a number of numerical experiments on benchmark datasets, which corroborate our findings." However, I do not think that the numerical results corroborate the theoretical results.

**Questions:**

**Critical Batchsize:** According to [1], the number of training steps is halved by each doubling of the batch size and that diminishing returns exist beyond a critical batch size. Is the definition of "the critical batchsize" used in the paper the same as in [1]? I cannot understand why "the critical batchsizes" are $b^* = \frac{\gamma}{\frac{1}{n} \Vert X \Vert^2}$ (linear least squares case; Page 4) and $b^* = \frac{4 \gamma + \frac{50 \gamma L^2 r^2}{\alpha}}{\Vert F(w_0) \Vert^2 + L^2 r^2}$ (nonlinear least squares case; Page 5).
Please give mathematically the definition of "the critical batchsize" and show that "the critical batchsizes" are $b^*$ (Pages 4 and 5).

[1] Christopher J Shallue, Jaehoon Lee, Joseph Antognini, Jascha Sohl-Dickstein, Roy Frostig, and
George E Dahl. Measuring the effects of data parallelism on neural network training. arXiv
preprint arXiv:1811.03600, 2018.

**Assumptions and numerical results:**
This paper gives sufficient conditions to satisfy Assumptions 1 and 2. I understand the authors try to give such sufficient conditions. However, we need to set $r$ (see (4.2)), $m$ (see (4.2)), $\beta$ (see (4.3)), and $\eta$ before implementing SGD using a constant step-size $\eta = \frac{1}{\beta}$. There is no how to set such parameters in Section 5 (Experimental results). Could the authors show evidences to emphasize that the parameters in Section 5 are based on Theorems 4.2-4.4?

**Numerical results:**
I think that the neural networks in Section 5 are small. I would like to check the performance of SGD for training wide-resnets on CIFAR-100 and ImageNet. I also would like to suggest that
- the authors plot the number of iterations to achieve high **test**/**training** accuracy versus the batch size and show the existence of critical batchsizes.

Moreover, I would like to suggest that
- the authors compare not only SGD using a constant step-size (e.g., $\eta$ is chosen on the basis of a grid search of $\eta \in (10^{-1}, 10^{-2}, 10^{-3} )$) but also the existing optimizers (e.g., momentum, Adam, and variants of Adam) with SGD using $\eta = \frac{1}{\beta}$.
If SGD using $\eta = \frac{1}{\beta}$ performs better than other optimizers,  SGD using $\eta = \frac{1}{\beta}$ is a good way to train deep neural networks.

**Optimization problem:**
This paper considers Problem (2.1) that is to minimize MSE: $\mathcal{L}(w) = \frac{1}{2n} \sum_{i=1}^n (f_i(w) - y_i)^2$. This setting seems to be limited to consider "a broader class of nonconvex problem" (see Abstract). Can we consider a more general nonconvex optimization, such as: Minimize $\mathcal{L}(w) = \frac{1}{n} \sum_{i=1}^n f_i (w)$, where $f_i$ is nonconvex and differentiable?

**Iteration complexity:**
The authors use "iteration complexity." Please define mathematically the iteration complexity used in the paper. Is the definition of the iteration complexity used in the paper the same as the definition in [2]?

[2] Yossi Arjevani, Yair Carmon, John C. Duchi, Dylan J. Foster, Nathan Srebro, and Blake Woodworth. Lower bounds for non-convex stochastic optimization. Mathematical Programming, 199(1):165– 214, 2023.

**Convergence speed:**
The abstract says that the paper gives a convergence speed of SGD. Please give a rate of convergence of SGD under Assumptions 1 and 2.

**Mathematical Preliminaries:**
The mathematical preliminaries are insufficient. For example,
- Theorem 1.1: $\lambda_{\min}$, $c$, and $\Vert \cdot \Vert_{\mathrm{op}}$ are not defined.
- (2.2): $\mathbb{E}_S$ is not defined.
- Line -8; Page 2: We cannot use $m$, since $m$ has been used to define the width $m$ (see Theorem 1.1).
- Line +2; Page 3: $S$ has been used to define the batch size $|S|$. The same notation is not good.
- Theorem 4.2 (4.1): $f$ is not defined.
- ...

**Proofs:**
Please give the proofs of Theorems 4.1--4.4 in the main body (I think that the proofs of Lemma 3.1, Theorem 3.2, Theorem 3.3 can move to the Appendix section. Then, the authors have sufficient space of the manuscript).  I do not believe that Theorems 4.1--4.4 are true, since there is no proof of theorems.

**Typos:**
- Theorem 1.1: where ... "and $m$ is the width of the network" is deleted.
- Figure 1: in each. --> in each
- Line -1; Page 7: opt
- ...

**My recommendation:** The paper includes theoretically interesting results. However, I have some concerns (the critical batchsize, the assumptions, the implementation of SGD, and numerical results). Hence, at this point, my score is 1: strong reject. However, I strongly believe that the authors could address my concerns. Hence, I hope to raise my score.

---

### Official Review · Reviewer_UrqU · 2023-11-04

**Soundness:** 3 good
**Presentation:** 4 excellent
**Contribution:** 2 fair
**Rating:** 5
**Confidence:** 3

**Summary:**

The paper rewrites a recent result on the linear convergence of SGD for wide neural networks to make explicit the batch size dependence of the rate. This allows the authors to exhibit the existence of a critical batch size for wide neural networks.

**Strengths:**

The paper is clear and to the point. It constitutes a first step in understanding the existence of a critical batch size for neural networks observed in some empirical studies.

**Weaknesses:**

- *Slightly limited result*: The paper's contribution is limited to theorem 3.3, a bound on the stochastic gradient norm for non-linear least squares. Almost the entirety of the rest comes from [1], especially from [1] section 3. Although the authors rightfully cite this work, they could have simply plugged in the $\beta$ they computed in Theorem 2.1 from [1] without restating several of the results and discussions from [1].

- *Wide neural networks vs linear models*: Theorem 3.3 is only worthwhile when the smoothness constant $L$ of the non-linearity very small (i.e so small that the problem is effectively linear) and when there is a solution achieving zero error not too far away. It is I believe clear now that wide neural networks are effectively linear models with interpolating solutions close to initialization so this work is essentially restating the result of [2] for linear least squares with small error terms that scale with $O(1/\sqrt{m})$. My point here is that this result is still a result for linear least squares and not for "a broader class of nonconvex problems" as stated in the abstract. This criticism is valid for all papers on wide neural networks but, for this work, since there is very little difficulty in dealing with the approximate linearity of wide nets, it is worth restating here.
---
[1] Liu, Chaoyue, et al. "Aiming towards the minimizers: fast convergence of SGD for overparametrized problems." arXiv preprint arXiv:2306.02601 (2023).

[2] Ma, Siyuan, Raef Bassily, and Mikhail Belkin. "The power of interpolation: Understanding the effectiveness of SGD in modern over-parametrized learning." International Conference on Machine Learning. PMLR, 2018.

**Questions:**

- On the convergence bound: The result provided is a high probability result that needs the existence of a solution in a ball whose radius scales with the failure probability. Are you choosing $m$ after fixing the failure probabilities to ensure the existence of such a solution ? Also I believe $\delta_1 \in (0, 1/5)$ and not of $1/3$ ?
- Why not simply reduce the moment bound assumption (3.1) to simply $\gamma$ bounded $x_i$ ? I have difficulty seeing any other scenario where this assumption is naturally shown to hold without assuming bounded x_i's.
- On the experiments: You choose to fix the same step size for all the batch sizes to illustrate the existence of a critical batch size. Could the authors explain why they make this choice especially since the empirical papers they cite studying critical batch sizes carefully note in the abstract that "how batch size affects model quality can largely be explained by differences in metaparameter tuning"(Shallue et al. (2018)) (and more importantly since Theorem 2.1 has a stepsize depending on $\beta$).
- Can you give more details on what is plotted in figure 2 column 2. Is the predicted rate obtained from computing the first terms of (4.3) on the datasets ? In addition, the range of values plotted is very small (from 3e-4 to 4e-4 for example on MNIST) so is the effect of batch size on the rate actually very small in practice ?
- I believe $w$ should be $w_0$ whenever you give expressions for $b*$ and $\beta$, so if my understanding is correct it is possible to simply compute the optimal batch size afterwhich there is diminished returns, could you tell us what they are for MNIST for example ?
- It doesn't appear that NanoGPT is converging linearly nor does it look like it's approaching zero training error, so I am unsure it fits in your setting of wide neural networks.

---

### Official Review · Reviewer_vAXR · 2023-11-04

**Soundness:** 2 fair
**Presentation:** 2 fair
**Contribution:** 2 fair
**Rating:** 5
**Confidence:** 3

**Summary:**

This paper studies the large-batch SGD training of neural networks. It argues that, under several assumptions, the convergence rate of mini-batch SGD for a network with large width, scales linearly to the batch size up to a constant which only depends on the initialization and training dataset.

**Strengths:**

This paper is well-motivated: the large-batch training problem is an important topic especially in the era of large model. The idea of relating the second moments of gradient distances with network width and then applying existing convergence analysis is interesting and promising.

**Weaknesses:**

The overall idea makes sense, but the paper in its current form is a bit hard to understand, partially due to missing important steps/details,  containing several mistakes, and lacking connection of the analysis with testing performance.

- How is the main technical result (Theorems 4.2) arrived: The authors first wrote "Theorems 4.2 and 4.2 directly imply...". Did you mean "Theorem 4.2 and Theorem 4.3"? Next, it is claimed that " A direct application of Theorem 4.1 yields the main result of the section", without any proof. I probably miss something here, but I did not see how the direct application works. In particular, how does the upper bound with linear forms in Theorem 4.1 transfer to the bound taking maximum form in Theorem 4.2?

- redefinition of S: The notation of S is overloaded. On page 2, S is defined as "a batch of indices". On page 3, however, S "denote the set of interpolating solutions". This makes it hard to readers to understand many things.

- mismatch of analytical scaling rate and observed rate empirically : An important motivation of this paper is, in the authors' own words, "explain why the performance of minibatch SGD with a large stepsize saturates". The offered claim (Theorem 1.1) gives a critical batch size and suggests that smaller batch size leads to linear scaling and larger batch size causes poor scaling. The empirical study sometimes violates this claim. For example, in Figure 2(f), the predicted convergence rate from the analytical form is far away from the true rate. Do you have any thoughts on why this is the case?

- lack of testing loss/accuracy: Neural networks are desired mostly due to their excellent generalization performance. How does the testing accuracy/loss change as the batch size increases? Does the convergence rate bound offers any insights on the generalization performance?

- performance as a function of time: The empirical analysis mostly focuses on training loss/error as a function of number iterations. How does it depend on the wall-clock time? After, mitigating the overall runtime is the goal of large batch training.

**Questions:**

Please see my comments above.

---

### Official Review · Reviewer_daaT · 2023-11-04

**Soundness:** 2 fair
**Presentation:** 2 fair
**Contribution:** 1 poor
**Rating:** 3
**Confidence:** 5

**Summary:**

This paper investigates the relationship between the batch-size and convergence speed for a broader class of nonconvex
problems. Unfortunately, the results are not novel, the introduction is very short it isn not clear the main contribution of the paper, and the comparison with related work is quite brief without much in-depth discussion.

**Strengths:**

N/A

**Weaknesses:**

The submission is unfortunately not very strong. The main results (Theorem 1.1) shows that the effective noise level reduces with batch size, which seems trivial. Indeed, I cannot see much novelty and intuition provided by Theorem 1.1.

---------------------

The introduction is very short it isn not clear the main contribution of the paper, and the comparison with related work is quite brief without much in-depth discussion.

---------------------

The paper considers a narrow optimization problem, and the setting in Theorem 1.1 is also quite limited.

Therefore, I do recommend rejection.

**Questions:**

Please see above.

---

### Meta-Review · Area_Chair_WJuM · 2023-12-10

**Metareview:**

The paper considers the convergence rate of SGD under changing batch size, in the standard non-convex setting. The relationship with interpolation (such as in sufficiently wide networks) is mentioned, which is interesting.
Unfortunately consensus among the reviewers is that it remains below the bar, and concerns remained on the theoretical formalism, key assumptions, and main claims. In particular the similarity to existing results - such as from least-squares - needs to be addressed. Unfortunately no author feedback was provided.

We hope the detailed feedback helps to strengthen the paper for a future occasion.

**Justification For Why Not Higher Score:**

all 4 reviews vote for reject

**Justification For Why Not Lower Score:**

N/A

---

### Decision · Program_Chairs · 2024-01-16

Reject